# Isoflavonoid Profiling and Estrogen-Like Activity of Four *Genista* Species from the Greek Flora

**DOI:** 10.3390/molecules25235507

**Published:** 2020-11-24

**Authors:** Antigoni Cheilari, Argyro Vontzalidou, Maria Makropoulou, Aggeliki K. Meligova, Nikolas Fokialakis, Sofia Mitakou, Michael N. Alexis, Nektarios Aligiannis

**Affiliations:** 1Department of Pharmacognosy and Natural Products Chemistry, Faculty of Pharmacy, National and Kapodistrian University of Athens, Panepistimiopoli Zografou, 15771 Athens, Greece; cheilarianti@pharm.uoa.gr (A.C.); avontzalidou@gmail.com (A.V.); mmakrop@pharm.uoa.gr (M.M.); fokialakis@pharm.uoa.gr (N.F.); mitakou@pharm.uoa.gr (S.M.); 2Institute of Chemical Biology, National Hellenic Research Foundation, 48, Vassileos Constantinou Avenue, 11635 Athens, Greece; ameligova@gmail.com

**Keywords:** *Genista depressa*, *Genista acanthoclada*, *Genista millii*, *Genista hassertiana*, isoflavones, estrogen-like activity

## Abstract

As part of our ongoing research on phytoestrogens, we investigated the phytochemical profile and estrogen-like activities of eight extracts from the aerial parts of four *Genista* species of Greek flora using estrogen-responsive cell lines. Ethyl acetate and methanolic extracts of *G. acanthoclada*, *G. depressa,*
*G. hassertiana,* and *G. millii* were obtained with accelerated solvent extraction and their phytochemical profiles were compared using ultra-high-performance liquid chromatography-high-resolution mass spectrometry (uHPLC-HRMS). Fourteen isoflavonoids, previously isolated from *G. halacsyi*, were used as reference standards for their identification in the extracts. Thirteen isoflavonoids were detected in both extracts of *G. acanthoclada* and *G. hassertiana*, while fewer and far fewer were detected in *G. millii* and *G. depressa,* respectively. The ethyl acetate extracts of *G. hassertiana* and *G. acanthoclada* displayed 2.45- and 1.79-fold higher, respectively, estrogen-like agonist activity in Ishikawa cells compared to MCF-7 cells at pharmacologically relevant concentrations. Both these extracts, but not that of *G. depressa*, contained mono- and di-*O-*β-d-glucosides of genistein as well as the aglycone, all three of which are known to display full estrogen-like activity at lower-than-micromolar concentrations. The possibility of using preparations rich in *G. hassertiana* and/or *G. acanthoclada* extracts as a potentially safer substitute for low-dose vaginal estrogen for menopausal symptoms is discussed.

## 1. Introduction

The *Genista* genus (Fabaceae) consists of about 100 species worldwide with ca. 18 species described in the flora of Greece [1]. The plants are usually small herbs and shrubs with an aromatic and bitter taste, with many of them being of economic interest, especially in the Mediterranean basin, due to their use as dyes, in animal feed market and in phytotherapy, with several of them reported having significant antioxidant capacity [2,3]. The *Genista* species are rich in flavonoids and especially isoflavonoids with estrogen-like activities of potential clinical importance [4,5]. Due to their structural similarity to estradiol, many isoflavone aglycones are able to interact with both isotypes of estrogen receptor (ER), ERα and ERβ [6,7]. The estrogen-like activities of isoflavones render them potentially therapeutic agents for the prevention and/or treatment of health issues due to estrogen deficiency, including increased risk for cardiovascular disease, menopausal symptoms, and post-menopausal osteoporosis [5,8,9]. Based on our ongoing effort in highlighting the biodiversity of Greek flora as a unique source of bioactive agents [10,11,12] and especially on our previous research in identifying isoflavonoids with estrogen-like activity from *G. halacsyi* [10], we selected four *Genista* species of Greek flora, namely *G. acanthoclada*, *G. depressa*, *G. hassertiana,* and *G. millii*, for phytochemical and biological evaluation. We report herein for the first time a comparative phytochemical characterization of the four *Genista* species, utilizing a uHPLC-HRMS identification approach, and an evaluation of their estrogen-like activity, as assessed using human endometrial and breast adenocarcinoma cells.

## 2. Results and Discussion

The present study reports the comparison of the phytochemical profile of the extracts from the aerial parts of *G. acanthoclada, G. depressa, G. hassertiana*, and *G. millii*, and the evaluation of the estrogen-like activity of their ethylacetate (EtOAc) and methanol (MeOH) extracts. These were obtained using accelerated solvent extraction with the following yields: For the EtOAc and MeOH extract of G. *acanthoclada,* 2.9 and 19.8%, respectively, for *G. hassertiana*, 2.9 and 15.3%, respectively, for *G. depressa*, 3.6 and 8.3%, respectively, and for *G. millii*, 1.4 and 19.1%, respectively. Notably, the yields of the EtOAc extract of *G. millii* and the MeOH extract of *G. depressa* were nearly half as much as those of the other three respective extracts, implying that substantial differences may exist in the phytochemical composition of aerial parts of *G. depressa* and *G. millii* compared to those of G. *acanthoclada* and *G. hassertiana*.

Regarding the phytochemical composition of the extracts, a preliminary thin layer chromatography (TLC) and high-performance liquid chromatography-ultraviolet (HPLC-UV) profile screening indicated that several qualitative and quantitative differences exist between the very many secondary metabolites present in each of these extracts. Hence, uHPLC-HRMS was employed for profiling and systematic identification of isoflavonoids in the extracts. HPLC-MS analysis was selected over HPLC-UV, since fragmentation mass spectra (MS/MS) could provide high confidence through the comparison to our previously isolated standards. The experiments were performed in positive and negative ion-mode, but most of the structural information derived from full-scan negative ion-mode. Having as reference standards the compounds isolated in our previous study of *G. halacsyi* [10], we confirmed the presence of two diglucosides of genistein [8-*C*,4′-*O*-di-glucopyranosyl genistein (**1**) and 7,4′-di-*O*-glucopyranosyl genistein (**2**)], five isoflavone monoglucosides [8-*C*-glucopyranosyl orobol (**3**), 7-*O*-glucopyranosyl isoprunetin (**4**), 8-*C*-glucopyranosyl 3′-*O*-methylorobol (**5**), 7-*O*-β-d-glucopyranosyl genistein (**6**), and 8-*C*-glucopyranosyl genistein (**7**)], as well as seven isoflavonoid aglycones [daidzein (**8**), genistein (**9**), isoprunetin (**10**), 5-*O*-methylorobol (**11**), 8-methoxyformononetin (**12**), 3′-methoxyisoprunetin (**13**), and biochanin A (**14**)] in some or all of the eight extracts (Table 1 and Appendix A). Of note, the isoflavonoids of Table 1 were previously isolated from a conventional MeOH extract of the aerial parts of *G. halacsyi*, using fast centrifugal partition chromatography followed by prep-TLC and/or silica and/or sephadex column chromatography of the fractions to deliver isolated compounds (>95% purity), as assessed by several spectrometric methods [10]. The data of Table 1 show that isoflavonoids **4**, **5**, **8**, **9**, **11**, **13**, and **14** were detected in all four *Genista* species, while **12** was not found in G. *acanthoclada* and *G. hassertiana*, **2** and **10** were not detected in *G. depressa* and *G. millii,* and **1**, **3**, **6**, and **7** were not observed in *G. depressa*. We have reported that **2**, **6**, and **7**, are the main constituents of the aerial parts of *G. halacsyi* [10]. The limited phytochemical compositions of the aerial parts of *G. millii* and, in particular, of *G. depressa*, with **2** missing from the first and **6** missing from both, compared to the respective compositions of G. *acanthoclada* and *G. hassertiana* are possibly related to the comparatively lower yields of the EtOAc extract of *G. millii* and the MeOH extract of *G. depressa*. Regarding the phytochemical composition of the MeOH extracts, in particular, only six isoflavonoids are featuring in the MeOH extract of *G. depressa* as compared to ten in that of *G. millii* and to thirteen in those of G. *acanthoclada* and *G. hassertiana*. We have previously reported that at the concentration of 1 μM, **2** and **6** displayed similar and higher, respectively, estrogen-like activity in Ishikawa cells compared to genistein, compatible with full conversion to the aglycone by cell membrane glucosidases, while **7** exhibited weak activity at this concentration, reflecting its ability to enter the cell using glucose transporters in the cell membrane as well as its metabolic stability [10]. Given that **2** is absent from *G. millii* extracts and **2** and **6** from *G. depressa* extracts, one may expect substantial differences in their estrogen-like activities compared to those of G. *acanthoclada* and *G. hassertiana*.

All eight extracts were evaluated, (i) for estrogen agonist activity by assessing their efficacy in inducing alkaline phosphatase (AlkP) expression in estrogen-free Ishikawa cells, using 0.1 nM estradiol i.e., the average post-menopausal concentration of the hormone in women’s serum, as a positive control and (ii) for estrogen antagonist activity by assessing their efficacy in suppressing AlkP expression in estrogen-free Ishikawa cells repleted with 0.1 nM estradiol, using 0.1 μM ICI182,780 as a positive control. Induction of AlkP activity in Ishikawa cells was previously found to be a reliable measure of estrogenic activity [6,13] and to depend on ERβ as well as ERα [14]. Τhe extracts were also evaluated for stimulatory and suppressive effects on the proliferation of estrogen-free and estradiol-repleted MCF-7 cells, respectively. Unlike Ishikawa cells which are known to express both ERα and ERβ, MCF-7 cells are known to express only ERα. Both cell lines have been widely used for the discovery and/or detection of natural or synthetic agents with estrogen-like activity [6,10,11,12,13,14,15]. The estrogen-like agonist effects of the extracts on the AlkP expression in Ishikawa cells and the proliferation of MCF-7 cells are shown in Table 2 and their antagonist effects in Appendix A. It is worth noting that none of the eight extracts displayed estrogen-antagonist activity in the concentrations tested.

Table 2 shows that six of the eight extracts, when tested at the concentration of 1 μg/mL, displayed full estrogen-like agonist activity in Ishikawa cells, while the EtOAc extracts of *G. millii* and *G. depressa* displayed partial and weak agonist activity, respectively. However, when tested at the concentration of 0.1 or 0.01 μg/mL, all eight extracts displayed weak or marginal agonist activity. Table 2 also shows that of the six extracts with full estrogen agonist activity in Ishikawa cells, three exhibited full estrogen agonist activity in MCF-7 cells as well, while two (the EtOAc extract of *G. acanthoclada* and the MeOH extract of *G. Depressa*) displayed partial and one (the EtOAc extract of *G. hassertiana*) weak estrogen-like agonist activity in MCF-7 cells. Specifically, Table 2 shows that the EtOAc extract of *G. hassertiana*, the EtOAc extract of *G. acanthoclada*, and the MeOH extract of *G. depressa* displayed 2.45-, 1.79-, and 1.52-fold higher, respectively, estrogen-like agonist activity in Ishikawa cells compared to MCF-7. These findings may be taken to suggest that the EtOAc extract of *G. hassertiana*, and to a lesser extent the other two as well, may combine a rather satisfactory breast safety profile with efficacy to prevent the genitourinary syndrome of menopause (GSM), as already proposed for other isoflavonoid-rich extracts [6].

We have previously reported that the MeOH extract of *G. halacsyi* is very rich in 7,4′-di-*O*-glucopyranosyl genistein (**2**) and 7-*O*-β-d-glucopyranosyl genistein (**6**); and that **6** and genistein displayed 1.59- and 1.24-fold higher, respectively, estrogen-like agonist activity in Ishikawa cells compared to MCF-7 cells, while **2** displayed practically the same estrogen-like agonist activity in both cells [10]. These findings could be taken to indicate that the higher estrogen-like activity of the EtOAc extracts of *G. hassertiana* and *G. acanthoclada* and the MeOH extract of *G. depressa* in Ishikawa cells vs. MCF-7 cells could be primarily attributed to genistein and **6**. However, the estrogen-like activities of isoflavonoids in extracts are expected to differ from their activities in pure form. Indeed, it has been reported that the ER-dependent transcriptional activity of isoflavones in pure form is different from that in mixture with other components [16]; and that cell uptake of isoflavone aglycones may be affected by companion isoflavonoids interfering with the hydrolysis of their respective glucosides by membrane glucosidases [17]. Nonetheless, genistein and its glucosides are likely to predominantly determine the estrogenic-like activities of the extracts, given that the ER-binding affinity and estrogen-agonist activity of genistein are much higher than those of the other isoflavones detected in the eight extracts [10].

We concluded above that the estrogen-like activities of the extracts are expected to predominantly reflect the activity of genistein and its glucosides as modulated by the remainder isoflavonoids and the unidentified components of the extracts. In line with this notion, we have previously discussed the possibility of using preparations rich in **6** and genistein to substitute for estrogen in the low-dose vaginal estrogen formulations prescribed for GSM [10,18]. However, while the data of Table 1 in conjunction with those of Table 2 suggest that the estrogen-like activity of all eight extracts could be primarily attributed to their isoflavonoid content, the uHPLC-HRMS analysis indicated that these extracts also contain several unidentified secondary metabolites. These warrant identification and biological evaluation in order to determine whether and how they might modulate the estrogen-like activity of the extracts in Ishikawa cells vs. MCF-7 cells.

The GSM is characterized by vaginal atrophy and dryness, causing dyspareunia, and/or by urinary tract infections, causing dysuria [19]. A low-dose estradiol vaginal cream can reportedly restore vaginal cytology, alleviate GSM, and restore the quality of life after menopause [18]. Although using vaginal estrogen to treat GSM is associated with a low risk to develop breast or endometrial adenocarcinoma, women’s compliance with hormonal interventions remains low, while their preference for herbal remedies compared to drugs remains high [20]. Interestingly, it has been reported that a vaginal gel containing isoflavones may relieve GSM as effectively as vaginal estrogen [21,22]. Given that estrogen ablation therapies of endocrine-related cancer are strongly associated with GSM [23], preparations of isoflavonoids for vaginal application may constitute a safer alternative to low-dose vaginal estrogen, provided that systemic absorption of the preparation’s active ingredients is deemed limited. Our findings indicate that the EtOAc extracts of *G. hassertiana* and *G. acanthoclada* warrant investigation of their potential to safely treat GSM using the ovariectomized mouse model of menopause.

## 3. Materials and Methods

### 3.1. General Experimental Procedures

Analytical TLC was performed on silica gel plates. Visualization of the TLC plates was achieved under UV at 254 and 366 nm as well as spraying with a 1:1 mixture of 5% H_2_SO_4_ and 5% vanillin in MeOH followed by heating. Evaporation of solvents was carried out on a vacuum rotary evaporator (Rotavapor R-3000r, Buchi, Switzerland). The optical densities in the biological assays were recorded using a *Tecan Safire II* microplate reader and Magellan 5.02 software.

### 3.2. Bio-Chemicals and Reagents

All solvents and reagents for phytochemical analyses were purchased from Sigma–Aldrich and Merck. Ishikawa human endometrial adenocarcinoma cells were purchased from ECACC and MCF-7 cells from ATCC. Phenol-red-free Minimal Essential Medium (MEM), penicillin, streptomycin, crystal violet, and p-nitrophenyl phosphate (pNPP) were obtained from Sigma-Aldrich, estradiol from Steraloids, and fetal bovine serum (FBS) from Biosera. Dextran-coated-charcoal-treated FBS (DCC-FBS) was treated to remove endogenous steroids as already described [24]. The ER degrader ICI182,780 was purchased from Tocris Bioscience and insulin from Biochrom.

### 3.3. Plant Material

*Genista acanthoclada* DC. was collected on the campus of the Faculty of Pharmacy in Athens, Attica. *Genista hassertiana* Bald. Buchegger (endemic of Greece) was collected in Mount Vourinos near Kozani. *Genista halacsyi* Heldr. (endemic of Greece) was collected from Mount Parnon, Peloponnese. *Genista millii* Heldr. Ex Boiss. (endemic of Greece) was collected from Oeti mountain. *Genista depressa* M. Bieb. was collected from Smolikas Mountain in the northern part of Ioannina. The aerial parts of the plant species were collected and identified by Dr. Eleftherios Kalpoutzakis. Voucher specimens thereof (NEK026, Bour005, KL121, ITI002, SM012) have been deposited in the herbarium of the Department of Pharmacognosy and Natural Products Chemistry, Faculty of Pharmacy, National and Kapodistrian University of Athens.

### 3.4. Preparation of Extracts

The dried pulverized aerial parts of all species were extracted successively in an accelerated solvent extraction (ASE 300—Dionex) apparatus with EtOAc and MeOH. 20 g of each plant was placed in stainless steel cells of 100 mL volume. Each solvent extraction was repeated two times and the obtained extracts were concentrated under reduced pressure at 40 °C. The extraction parameters were as follows; pressure 1500 psi, temperature 70 °C, pre-heating time 10 min, heating time 1 min.

### 3.5. Isoflavonoid Profiling and Identification

LC-MS experiments were performed on an Accela High-Speed LC System equipped with a photo diode array (PDA) detector and hyphenated to an LTQ-Orbitrap XL hybrid mass spectrometer, using an Electrospray ionization (ESI) ionization probe, in positive and negative mode (Thermo Scientific, Steingrund, Germany). Samples were injected at a concentration of 100 ppm for extracts and 10 ppm for standards, diluted in MeOH/H2O 50:50 in an Ascentis C18 150 by 2.1 mm, 3 μm column (Supelco Analytical, Bellefonte, PA) and a flow rate of 0.40 mL/min was used for the elution. The mobile phase consisted of solvents A: 0.1% Formic acid in water and B: Acetonitrile. A gradient method (total run time of 24 min with 5 min equilibration time) was used for the profiling of the samples as follows: 0 to 3 min: 5% B, 3 to 24 min: from 5% B to 95% B. Column temperature was kept at 30 °C throughout all experiments and the injection volume was 5 μL. MS parameters: Spray voltage: 4.5 kV, 350 °C; drying gas: 8 L/min; nebulizer: 30 psi; full scan mode: m/z 100–1500. Analysis was performed using the Fourier transform mass spectrometry mode of the LTQ orbitrap (FTMS) in the full scan ion mode, applying a resolution of 60,000. Mass spectra were processed with MZmine 2.32 [25] with mass detector centroid noise set at 1.0E3 with a minimum scan span of 5 continuous points above noise level.

### 3.6. Biological Assays

#### 3.6.1. Cell Cultures

Ishikawa human endometrial adenocarcinoma cells were cultured in phenol-red-free MEM supplemented with 100 U/mL penicillin, 100 μg/mL streptomycin, and 5% FBS. MCF-7 human breast adenocarcinoma cells were cultured in phenol-red-free MEM supplemented with 100 U/mL penicillin, 100 μg/mL streptomycin, 10% FBS, 1 mg/lt insulin, and 0.1 nM estradiol. Cell cultures were kept subconfluent at 37 °C in a humidified air chamber containing 5% CO_2_.

#### 3.6.2. Assessment of MCF-7 Cell Proliferation

Cells were plated in 96 well flat-bottom plates at a density of 4 × 10^3^ cells per well in phenol-red-free MEM supplemented with 5% DCC-FBS and allowed to settle for 24 h prior to assessing effects of the test extracts on cell viability using Trypan blue as already described [26]. Settled viable cells were fed with fresh medium supplemented with 0.1 nM estradiol, in order to determine estrogen antagonist activity, or with vehicle (0.1% dimethyl sulfoxide (DMSO)), in order to determine estrogen agonist activity, and were allowed to proliferate in the presence of test extracts (1 μg/mL) for 6 days, with a change to fresh medium at the 3rd day. Relative numbers of viable cells were determined using crystal violet as already described [27]. Optical densities were measured at 550 nm using a *Safire II* microplate reader (Tecan). Cells exposed to vehicle or to 0.1 nM estradiol (full estrogen agonist) in the absence or presence of 0.1 μM ICI182,780 (full estrogen antagonist) served as controls.

#### 3.6.3. Assessment of Alkaline Phosphatase Expression in Ishikawa Cells

Alkaline phosphatase expression in Ishikawa cells was assessed as already described [11]. Briefly, cells were plated in 96 well flat-bottom plates at a density of 12 × 10^3^ cells per well in phenol-red-free MEM supplemented with 5% DCC-FBS and were allowed to settle for 24 h prior to assessing effects of test extract on cell viability as already described [26]. Settled viable cells were fed with fresh medium supplemented with 0.1 nM estradiol or vehicle (0.1% DMSO) and then exposed to test extracts (1, 0.1, 0.01 μg/mL) for 3 days. AlkP activity was measured at 405 nm in a microplate reader (*Safire II*, Tecan) using pNPP. Cells exposed to vehicle or to 0.1 nM estradiol (full estrogen agonist) in the absence or presence of 0.1 μΜ ICI182,780 (full estrogen antagonist) served as controls.

## Figures and Tables

**Table 1 molecules-25-05507-t001:** uHPLC-HRMS-based identification of isoflavonoids in MeOH and EtOAc extracts of *G. acanthoclada*, *G. hassertiana*, *G. depressa*, and *G. millii*.

Peak	t*_R_* (min)	*m*/*z*	[M]^−^	M. Formula	Compound	*G. Acanthoclada*		*G. Hassertiana*		*G. Depressa*		*G. Millii*	
						*MeOH*	*EtOAc*	*MeOH*	*EtOAc*	*MeOH*	*EtOAc*	*MeOH*	*EtOAc*
1	6.96	593.150	[M–H]^−^	C_27_H_30_O_15_	8-*C*,4′-*O*-diglucopyranosyl genistein	yes	yes	yes	yes	N.D	N.D	yes	yes
2	7.04	593.150	[M–H]^−^	C_27_H_30_O_15_	7,4′-di-*O*-glucopyranosyl genistein	yes	yes	yes	yes	N.D	N.D	N.D	N.D
3	7.6	447.093	[M–H]^−^	C_21_H_20_O_10_	8-*C*-glucopyranosyl orobol	yes	yes	yes	yes	N.D	N.D	yes	yes
4	7.89	445.114	[M–H]^−^	C_22_H_22_ O_10_	7-*O*-glucopyranosyl isoprunetin	yes	yes	yes	yes	yes	yes	yes	yes
5	8.041	461.109	[M–H]^−^	C_22_ H_22_ O_11_	8-*C*-glucopyranosyl-3′-*O*-methylorobol	yes	yes	yes	yes	yes	yes	yes	yes
6	8.34	431.098	[M–H]^−^	C_21_ H_20_ O_10_	7-*O*-β-d-glucopyranosyl genistein	yes	yes	yes	yes	N.D	N.D	yes	yes
7	9.06	431.098	[M–H]^−^	C_21_H_20_ O_10_	8-*C*-glucopyranosyl genistein	yes	yes	yes	yes	N.D	N.D	yes	yes
8	10.7	253.050	[M–H]^−^	C_15_H_10_ O_4_	daidzein	yes	yes	yes	yes	yes	yes	yes	yes
9	12.1	269.045	[M–H]^−^	C_15_H_10_O_5_	genistein	yes	yes	yes	yes	yes	yes	yes	yes
10	12.4	283.060	[M–H]^−^	C_16_H_12_O_5_	isoprunetin	yes	yes	yes	yes	N.D	N.D	N.D	N.D
11	12.5	299.055	[M–H]^−^	C_16_H_12_O_6_	5-*O*-methyl orobol	yes	yes	yes	yes	yes	yes	yes	yes
12	13.7	297.076	[M–H]^−^	C_17_H_14_O_5_	8-methoxyformononetin	N.D	N.D	N.D	N.D	yes	yes	yes	yes
13	14.0	313.071	[M–H]^−^	C_17_H_14_O_6_	3′-methoxyisoprunetin	yes	yes	yes	yes	N.D	yes	N.D	yes
14	15.3	283.060	[M–H]^−^	C_16_H_12_O_5_	biochanin A	yes	yes	yes	yes	N.D	yes	N.D	yes

N.D. Not detected. **^a^** Compounds **1**–**14** have been previously isolated and identified from *G. halacsyi* [10].

**Table 2 molecules-25-05507-t002:** Estrogen agonist activity of *Genista* extracts.

	Alkaline Phosphatase Expression (Ishikawa Cells) Agonism ^a^ (% of Estradiol ^b^)			Cell Proliferation (MCF-7 Cells) Agonism ^a^ (% of Estradiol ^b^)
	Extract (1 μg/mL)	Extract (0.1 μg/mL)	Extract (0.01 μg/mL)	Extract (1 μg/mL)
**Estradiol ^b^**	100	100	100	100
***G. millii*** **—EtOAc**	63.9 ± 4.7 (P)	<10% (M)	M	F (87.6 ± 1.3)
***G. millii*** **—MeOH**	F (82.4 ± 5.6)	W (15.5 ± 3.5)	W (14.2 ± 7.4)	F (115.5 ± 16.6)
***G. acanthoclada*** **—EtOAc**	F (81.2 ± 6.3)	W (25.1 ± 5.3)	M	P (45.4 ± 5.8)
***G. acanthoclada*** **—MeOH**	F (84.6 ± 2.5)	W (16.0 ± 4.5)	M	F (71.6 ± 2.0)
***G. hassertiana*** **—EtOAc**	F (82.2 ± 3.0)	W (12.5 ± 0.1)	M	W (33.5 ± 8.1)
***G. hassertiana*** **—MeOH**	F (100.7 ± 9.8)	W (29.1 ± 6.6)	M	F (72.3 ± 10.1)
***G. depressa*** **—EtOAc**	W (32.4 ± 2.0)	M	M	N.D.
***G. depressa*** **—MeOH**	F (88.3 ± 6.0)	M	M	P (58.2 ± 9.8)

^a^ % Agonism = (OD_extract_ − OD_vehicle_)*100/(OD_estradiol_ − OD_vehicle_); OD: Optical density at 405 nm. Values are mean ± SEM of three independent experiments involving triplicate test points. Agonist effects were classified as full (F), partial (P), or weak (W) depending on whether induction of alkaline phosphatase expression and MCF-7 cell proliferation was ≥67%, 34–66% or 10–33%, respectively, of that of estradiol. Effects <10% were classified as marginal (M). ^b^ Estradiol was used at 0.1 nM; of note, the estrogen agonist activity of 0.1 nM E2 in Ishikawa cells and in MCF-7 cells is similar to that of 1 μM genistein [10,12]. N.D., not detected.

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
