# Peer review of "Isoflavonoid Profiling and Estrogen-Like Activity of Four Genista Species from the Greek Flora"

_molecules, 2020, doi:10.3390/molecules25235507_

Round 1

Reviewer 1 Report

 The extraction of  phytoestrogens, by different solvents and their phytochemical   profile were  investigated. The estrogen-like activities of eight extracts from the aerial parts of four Genista species of   Greek flora using estrogen-responsive cell lines was tested and reported. In particular isoflavones both aglycones and  glycosides.

The activity was tested  by  Ishikawa and  MCF-7 cells  and compared with genistein.

The point of the study is interesting and the data reported have some important points that , as the same AAs say  “

 The possibility of using preparations rich in G. 23 hassertiana and/or G. acanthoclada extracts as potentially safer substitute for low-dose vaginal  estrogen for menopausal symptoms is discussed.” B ut they conclude  that more investigation are required. But they say in conlusions  :

“Our findings indicate that the EtOAc extracts of G. hassertiana  and G. acanthoclada warrant investigation of their potential to safely treat GSM using the  ovariectomized mouse model of menopause”.

Thus it seems that more work i sto be done.

The absence of a defined discussion in conclusion of the paper  also denotes that these are relevant but very preliminary results.

The paper follows the research line  well performed and well documented in the paper of ref 10.

To my opinion in this case  the results are rather limited  with respect to the previous work.  

The impression is that the authors in this case  identify  the composition of the extracts only on the basis of the activity on cells taking genistein,-I suppose pure- as reference ( 100%). I do not know if  this procedure is completely  correct. I believe is that should be at least better  validated. The presence of other components ( i.e. dadzein as example) in the extract may interfere with the compounds?.

Moreover the AAs are really  sure that also in this context the activity of aglycones and glycosides are substantially the same asd already published for pure components (10)?  In fact in ref 10 the compounds seemed to be isolated  not using the simple extract . The differences in the protocols ( purity , isolations,  impurities if any)  should be indicated with respect to the results of ref 10.

These points should  be better detailed:

  1. The MS spectra are sufficient for determine the purity of the extract?.
  2. The UV spectra may help to document better these samples .
  3. The components absent “ not detected”  are attributed to low concentration , but which is the limit of detection of the procedure adopted?.
  4. The plant metabolism is never considered in the study , thus the term metabolic should be erased everywhere . On the contrary to my opinion the term profiling is used very correctly.
  5. In conclusion the results presented  should be  limited  in the form of a short letter ( or communication)  that may complement the previous papers in the field much more detailed. The  other possibilility is to implement with more experimental data the paper to obtain a good  article as in ref 10.

    Finally many sentences are copied and pasted from ref 10.This should be corrected. 

Author Response

Reviewer #1: The extraction of phytoestrogens, by different solvents and their phytochemical profile were investigated. The estrogen-like activities of eight extracts from the aerial parts of four Genista species of Greek flora using estrogen-responsive cell lines was tested and reported. In particular isoflavones both aglycones and glycosides. The activity was tested by Ishikawa and MCF-7 cells and compared with genistein.

Point 1: The point of the study is interesting and the data reported have some important points that, as the same AAs say,“The possibility of using preparations rich in G. hassertiana and/or G. acanthoclada extracts as potentially safer substitute for low-dose vaginal estrogen for menopausal symptoms is discussed.” But they conclude that more investigation are required. But they say in conlusions: “Our findings indicate that the EtOAc extracts of G. hassertiana and G. acanthoclada warrant investigation of their potential to safely treat GSM using the ovariectomized mouse model of menopause”. Thus it seems that more work i sto be done.

RESPONSE: The moto “Our findings…..warrant investigation” is extensively used in the Conclusion sections of research articles to denote that the findings indicate new research objectives. The last sentence of our ms reads as follows: “Our present findings indicate that the extracts of G. hassertiana and G. acanthoclada warrant investigation of their potential to safely treat GSM using the ovariectomized mouse model of menopause”. However, histopathological evaluation of the effect of Genista extracts on the vagina and uterus of ovariectomized mice (and ideally, of aged female mice as well) would have required inventing and standardizing an appropriate carrier of the extract (e.g. a cream) as well as a secure method to accurately apply the carried extract in the vagina of mice over some appreciable time span, eventually involving several repeated applications of the preparation in the vagina during this time span. In addition, one would have to determine the systemic absorption of the preparation’s active ingredients and to secure that any side effects to the breast and the uterus are limited. This is an intricate and sizeable research endeavor the undertaking of which would have conceptually derailed this report from its chemistry-oriented scope, which is set to fit the scope defined for the section Natural Products Chemistry; special issue Plant Natural Products.

Point 2: The absence of a defined discussion in conclusion of the paper also denotes that these are relevant but very preliminary results. The paper follows the research line well performed and well documented in the paper of ref 10. To my opinion in this case the results are rather limited with respect to the previous work.

RESPONSE: Our ms does not have a separate Conclusion section. However, the research described in the ms reached several interesting conclusions that are discussed separately. Specifically, the last four paragraphs of the Results and Discussion section have a final sentence that serves as the conclusion of what is discussed in the respective paragraph. Thus, the fourth paragraph from the end concludes that (lines 133-136): “These findings may be taken to suggest that the EtOAc extract of G. hassertiana, and to a lesser extent the other two as well, may combine a rather satisfactory breast safety profile with efficacy to prevent the genitourinary syndrome of menopause (GSM), as already proposed for other isoflavonoid-rich extracts [6].” The third paragraph from the end concludes that (lines 148-151): “Nonetheless, genistein and its glucosides are likely to predominantly determine the estrogenic-like activities of the extracts, given the higher ER-binding affinity and estrogen-agonist activity of genistein compared to the other isoflavones detected in the eight extracts [10].” The second paragraph from the end concludes that (lines 158-161): “…the uHPLC-HRMS analysis indicated that these extracts also contain several unidentified secondary metabolites. These warrant identification and biological evaluation in order to determine whether and how they might modulate the estrogen-like activity of the extracts in Ishikawa cells vs MCF-7 cells.”  Finally, the last paragraph concludes that (lines 172-174): “Our findings indicate that the EtOAc extracts of G. hassertiana and G. acanthoclada warrant investigation of their potential to safely treat GSM using the ovariectomized mouse model of menopause.”

Point 3: The impression is that the authors in this case identify the composition of the extracts only on the basis of the activity on cells taking genistein,-I suppose pure- as reference (100%). I do not know if this procedure is completely correct. I believe is that should be at least better validated. The presence of other components ( i.e. dadzein as example) in the extract may interfere with the compounds?.

RESPONSE: We used 0.1 nM estradiol rather than genistein as control because we deemed as more important to compare the estrogen-like activity of extracts to that of the average post-menopausal concentration of the hormone in women’s serum in view of potentially using the extracts as substitutes for low-dose vaginal estrogen. Of note, throughout the years, we constantly observed that the estrogen-like activity of 1 μΜ genistein is very similar to that of 0.1 nM estradiol in both Ishikawa and MCF-7 cells (please consult Table 2 and Figs 3A, 3B and 5A in reference [10]  and Table 2 and Figs 3C and 4B in reference [12] to confirm this point). In this light, the footnote (b) of Table 2 (lines 120-122) was revised to read as follows: “Estradiol was used at 0.1 nM; of note, the estrogen agonist activity of 0.1 nM E2 in Ishikawa cells and in MCF-7 cells is very similar to that of 1 μM genistein [10, 12].” Daidzein and other components are likely to contribute to the estrogen-like activity of the extracts. The reviewer is kindly asked to consult our response to point 4 to resolve this issue.

Point 4: Moreover the AAs are really sure that also in this context the activity of aglycones and glycosides are substantially the same asd already published for pure components (10)?

RESPONSE: To answer the above question the ms was revised to read as follows (lines 143-151): However, the estrogen-like activities of isoflavonoids in extracts are expected to differ from their activities in pure form. Indeed, it has been reported that the ER-dependent transcriptional activity of isoflavones in pure form is different from that in mixture with other components [16]; and that cell uptake of isoflavone aglycones may be affected by companion isoflavonoids interfering with the hydrolysis of the respective glucosides by membrane glucosidases [17]. Nonetheless, genistein and its glucosides are likely to predominantly determine the estrogenic-like activities of the extracts, given that the ER-binding affinity and estrogen-agonist activity of genistein are much higher than those of the other isoflavones detected in the eight extracts [10].

Point 5: In fact in ref 10 the compounds seemed to be isolated not using the simple extract. The differences in the protocols ( purity , isolations, impurities if any) should be indicated with respect to the results of ref 10.

RESPONSE: In response to the above request the ms was revised to read as follows (lines 74-77): “Of note, the isoflavonoids of Table 1 were previously isolated from a conventional MeOH extract of the aerial parts of G. halacsyi, using fast centrifugal partition chromatography followed by prep-TLC and/or silica and/or sephadex column chromatography of the fractions to deliver isolated compounds (>95% purity), as assessed by several spectrometric methods [10].”

Point 6: These points should be better detailed:

6.1: The MS spectra are sufficient for determine the purity of the extract? The UV spectra may help to document better these samples.

RESPONSE: Plant extracts are complex natural matrices comprising numerous compounds; hence the comment “determine the purity of the extract” makes no sense to us. We revised our ms (line 59-64) as follows: “Regarding the phytochemical composition of the extracts, a preliminary TLC and HPLC profile screening indicated that several qualitative and quantitative differences exist between the very many secondary metabolites present in each of these extracts. Hence, uHPLC-HRMS was employed for profiling and systematic identification of isoflavonoids in the extracts. HPLC-MS analysis was selected over HPLC-UV, since fragmentation mass spectra (MS/MS) could provide high confidence through the comparison to our previously isolated standards.”

6.2: The components absent “not detected” are attributed to low concentration, but which is the limit of detection of the procedure adopted?.

RESPONSE: Section 3.5 was revised to read as follows (lines 217-220): “Analysis was performed using the Fourier transform mass spectrometry mode of the LTQ orbitrap (FTMS) in the full scan ion mode, applying a resolution of 60,000. Mass spectra were processed with MZmine 2.32 [25] with mass detector centroid noise set at 1.0E3 with a minimum scan span of 5 continuous points above noise level.”

6.4: The plant metabolism is never considered in the study , thus the term metabolic should be erased everywhere. On the contrary to my opinion the term profiling is used very correctly.

RESPONSE: In the revised ms the word “profiling” substituted for the term “metabolic profiling”. The revised phrase reads as follows (lines 63-64): “Hence, uHPLC-HRMS was employed for profiling and systematic identification of isoflavonoids in the extracts.”

Point 7: In conclusion the results presented should be limited in the form of a short letter ( or communication) that may complement the previous papers in the field much more detailed. The other possibilility is to implement with more experimental data the paper to obtain a good article as in ref 10.

RESPONSE: According to the Instructions to Authors, “Molecules has no restrictions on the length of manuscripts, provided that the text is concise and comprehensive”. Our ms is concise but on the other hand too long to be published as a letter. Whether our ms could be published as regular article or communication is a decision of the Editor. As for the reviewer’s suggestion to avert publication of this work on the grounds that more data are required to “obtain a good article as in ref 10”, we object to his suggestion on the grounds that “quality (is) over quantity”, as the saying goes. It is up to the Editor to decide whether and in what form this ms could be published. To obtain more experimental data by subjecting the Genista extracts to the detailed biological evaluation of the pure isoflavonoids featuring in ref. 10 would be in disharmony with the predominantly chemistry-oriented scope set by Molecules for the section Natural Products Chemistry; special issue Plant Natural Products.

Point 8: Finally many sentences are copied and pasted from ref 10.This should be corrected.

RESPONSE: Using appropriate publicly available software we spotted the following four sentences, all in the M & M section, that might be considered the result of plagiarism.

  1. i) Values are Mean ± SEM of three independent experiments carried out in triplicate
  2. ii) Analytical TLC was carried out on silica gel plates.

iii) Evaporation of solvents was performed on a vacuum rotary evaporator (Rotavapor R-3000r, Buchi, Switzerland).

  1. iv) AlkP activity was assessed at 405 nm in a Safire II microplate reader using pNPP.

These are standard expressions used to describe routine activities. They were not identified as plagiarism when slightly rephrased in the revised ms to read as follows:

  1. i) Values are Mean ± SEM of three independent experiments involving triplicate test points. (lines 116-117)
  2. ii) Analytical TLC was performed on silica gel plates. (line 177)

iii) Evaporation of solvents was carried out on a vacuum rotary evaporator (Rotavapor R-3000r, Buchi, Switzerland). (lines 179-180)

  1. iv) AlkP activity was measured at 405 nm in a microplate reader (Safire II, Tecan) using pNPP. (lines 245-246)

We are grateful to all three reviewers for their invaluable help in improving the text structure and reading comprehensiveness of our original submission. Hopefully, they will find the above revisions satisfactory.

Sincerely

Reviewer 2 Report

It is a good paper on natural products - isoflavonoids profiling - and on the evaluation of the estrogen-like activity of extracts of four Genista species from the Greek flora.
Work well done and manuscript very well organised and written.
There are some small mistakes, since in some compound names, O- must be italicized and it is not.

Manuscript must be accepted after minor revision.

Author Response

Reviewer #2: It is a good paper on natural products - isoflavonoids profiling - and on the evaluation of the estrogen-like activity of extracts of four Genista species from the Greek flora. Work well done and manuscript very well organised and written.
There are some small mistakes, since in some compound names, O- must be italicized and it is not. Manuscript must be accepted after minor revision.

RESPONSE: We corrected the names of three compounds (two in the main text and one in the Supplement) as suggested by the reviewer.

We are grateful to all three reviewers for their invaluable help in improving the text structure and reading comprehensiveness of our original submission. Hopefully, they will find the above revisions satisfactory.

Sincerely

Reviewer 3 Report

The manuscript (molecules-996941) entitled “Isoflavonoid profiling and estrogen-like activity of 2 four Genista species from the Greek flora” is a manuscript by Antigoni Cheilari, et al. The authors studied the phytochemical profile and estrogen-like activities of eight extracts from the aerial parts of four Genista species of Greek flora using estrogen-responsive cell lines. ethyl acetate extracts of G. hassertiana and G. acanthoclada, but not that of G. depressa, contained mono- and di-O-β-D-glucosides of genistein as well as the aglycone, all three knowns to display full estrogen-like activity at lower-than-micromolar concentrations.

Major comments

  1. Some language editing is needed. In line 22 on page 1/9 (abstract section), “all three knowns…”? Please check.
  2. Some editing is needed. The first paragraph of the Result and Discussion section is too long. Please separate it into 3 to three paragraphs to shown the comparison of the EtOAc and MeOH extracts, identified molecules using uHPLC-HERM, and discussion of the components between the previous and current studies.
  3. In study shown in table 2, genistein probably should be included as a control.

Author Response

Reviewer #3: The manuscript (molecules-996941) entitled “Isoflavonoid profiling and estrogen-like activity of four Genista species from the Greek flora” is a manuscript by Antigoni Cheilari, et al. The authors studied the phytochemical profile and estrogen-like activities of eight extracts from the aerial parts of four Genista species of Greek flora using estrogen-responsive cell lines. ethyl acetate extracts of G. hassertiana and G. acanthoclada, but not that of G. depressa, contained mono- and di-O-β-D-glucosides of genistein as well as the aglycone, all three known to display full estrogen-like activity at lower-than-micromolar concentrations.

Point 1: Some language editing is needed. In line 22 on page 1/9 (abstract section), “all three knowns…”? Please check.

RESPONSE: Language editing of the revised ms was carried out using “grammarly” software. The penultimate sentence of the abstract was revised to read as follows (lines 21-23): “Both these extracts, but not that of G. depressa, contained mono- and di-O-β-D-glucosides of genistein as well as the aglycone, all three of which are known to display full estrogen-like activity at lower-than-micromolar concentrations.

Point 2: Some editing is needed. The first paragraph of the Result and Discussion section is too long. Please separate it into 3 to three paragraphs to shown the comparison of the EtOAc and MeOH extracts, identified molecules using uHPLC-HERM, and discussion of the components between the previous and current studies.

RESPONSE: In the revised ms the first paragraph of the Result and Discussion section was separated into three paragraphs as requested.

Point 3: In study shown in table 2, genistein probably should be included as a control.

RESPONSE: We used 0.1 nM estradiol rather than genistein as control because we deemed as more important to compare the estrogen-like activity of extracts to that of the average post-menopausal concentration of the hormone in women’s serum in view of potentially using the extracts as substitutes for low-dose vaginal estrogen. Of note, throughout the years, we constantly observed that the estrogen-like activity of 1 μΜ genistein in both Ishikawa and MCF-7 cells is very similar to that of 0.1 nM estradiol (please consult Table 2 and Figs 3A, 3B and 5A in reference [10] and Table 2 and Figs 3C and 4B in reference [12] to confirm this point). In this light, the footnote (b) of Table 2 was revised to read as follows (lines 120-122): “Estradiol was used at 0.1 nM; of note, the estrogen agonist activity of 0.1 nM E2 in Ishikawa cells and in MCF-7 cells is very similar to that of 1 μM genistein [10, 12].

We are grateful to all three reviewers for their invaluable help in improving the text structure and reading comprehensiveness of our original submission. Hopefully, they will find the above revisions satisfactory.

Sincerely

Round 2

Reviewer 1 Report

Now the MS is better modified. I have no comments.